# Three-dimensional motion analysis of pre- and postoperative thumb movement in trapeziometacarpal joint osteoarthritis— Comparison of arthrodesis and trapeziectomy with suspensionplasty

**Teruyasu Tanaka**[1], **Akira Kodama**[1]*, **Hiroshi Kurumadani**[2], **Kaguna Tanimoto**[1], **Shigeki Ishibashi**[1], **Masaru Munemori**[1], **Toru Sunagawa**[2], **Nobuo Adachi**[1]

1 Department of Orthopaedic Surgery, Graduate School of Biomedical and Health Sciences, Hiroshima University, Hiroshima, Japan, 2 Laboratory of Analysis and Control of Upper Extremity Function, Graduate School of Biomedical and Health Sciences, Hiroshima University, Hiroshima, Japan

* akirakodama@hiroshima-u.ac.jp

## Abstract

Trapeziometacarpal osteoarthritis (TMC-OA) reduces the range of motion (ROM) of the thumb. However, the kinematic change achieved through surgical treatment remains unclear. Therefore, to quantify the kinematic change following TMC-OA surgery, we performed a three-dimensional motion analysis of the thumb using an optical motion capture system preoperatively and 1 year postoperatively in 23 patients with TMC-OA scheduled for arthrodesis (AD) or trapeziectomy with suspensionplasty (TS). Eighteen hands of nine healthy volunteers were also included as controls. Both procedures improved postoperative pain and Disability of the Arm, Shoulder and Hand scores, and AD increased pinch strength. The ROM of the base of the thumb was preserved in AD, which was thought to be due to the appearance of compensatory movements of adjacent joints even if the ROM of the TMC joint was lost. TS did not improve ROM. Quantifying thumb kinematic changes following TMC-OA surgery can improve our understanding of TMC-OA treatment and help select surgical procedures and postoperative assessment.

## Introduction

Trapeziometacarpal osteoarthritis (TMC-OA) causes functional impairment of the upper extremities due to pain, deformity, and weakness. The range of motion (ROM) of the thumb is reduced in this condition, especially in the movements of extension and adduction [1]. The diminishment of manual dexterity can result in a decline in the activities of daily living [2]. The operative treatment for TMC-OA includes ligament reconstruction, arthrodesis, osteotomy, trapeziectomy with suspensionplasty, and prosthesis implementation to attain clinically satisfactory outcomes. It is widely acknowledged that the choice of surgical intervention is

**Data Availability Statement:** All relevant data are within the paper and its Supporting Information files.

**Funding:** The author(s) received no specific funding for this work.

**Competing interests:** The authors have declared that no competing interests exist.

often determined by the severity of the condition, the patient's needs, and the surgeon's preference. However, the definitive surgical procedure for the treatment of TMC-OA remains undetermined [3]. One reason for this is the intricate anatomy and associated motion of the thumb, which renders it unclear how each surgical procedure alters the mechanics of the TMC joint.

In a motion capture study on active oppositional movements using retroreflective markers on the thumb phalanx and metacarpal bones of the thumb and index finger, Haman et al. [4] showed that patients with TMC-OA had a reduced ROM at the TMC and metacarpophalangeal (MP) joints compared to healthy joints. Few studies have observed dynamic changes pre- and postoperatively. Kawano et al. [5] reported that arthrodesis (AD) reduces the thumb tip trajectory area by approximately 30% compared to that observed preoperatively in a cadaver study. Li et al. [6] reported a significant decrease in the angle of palmar and radial abduction following AD. Trapeziectomy with suspensionplasty (TS) is a commonly used procedure and is expected to maintain the TMC joint's ROM following surgery. However, Hatipoğlu et al. [7] reported satisfactory functional results after TS, but ROM was lower than the contralateral side. On the other hand, Wolf et al. [8] reported that the long-term outcome was comparable to the healthy contralateral side. Schröder et al. [9] evaluated the postoperative ROM of AD and TS using the distance between the interphalangeal (IP) joint and the second metacarpal head and reported no significant differences in motion between both procedures. Nevertheless, the actual effects of TMC surgery on joint movement and the degree of improvement in motor function and its mechanisms remain unclear. This is because most studies have compared postoperative kinematics with healthy joints, and few have compared them with preoperative OA joints, which would already have a reduced ROM.

This study aimed to quantify the postoperative kinematic changes between AD and TS in patients who have undergone surgery and clarify the characteristics of kinematic changes using each surgical procedure. In treating TMC-OA, AD is recognised for preserving the bone structure of the thumb, achieving a stable and pain-free joint. It is expected to improve grip and pinching strength compared to other surgical procedures. However, this advantage is counterbalanced by concerns about reducing the ROM and the precision of pinching movements. In contrast, TS is anticipated to preserve joint mobility, but there are concerns about the potential loss of pinching strength. Our hypotheses are (1) AD has little effect on ROM due to compensations in adjacent joints. (2) TS can improve ROM.

## Materials and methods

This prospective cohort study was conducted at a single institution. The Ethics Committee of the authors' institution approved the study (No. E-604). All participants signed the informed consent form approved by the review committee after being informed about the three-dimensional (3D) motion analysis study, the follow-up schedule, and the possible risks.

### Patients

A total of 23 patients out of 35 patients who underwent surgical treatment for TMC-OA at our institution from March 1, 2018 to June 30, 2021 were enrolled in this study. Seven individuals who underwent surgical procedures other than AD and TS, three who could not be followed, and two with insufficient data were excluded from the study. Patients with musculoskeletal diseases other than current TMC-OA (e.g., carpal tunnel syndrome), systemic inflammatory diseases, such as rheumatoid arthritis or gout, complex regional pain syndrome, or neurological diseases were excluded. Patients with TMC-OA caused by trauma and individuals who had undergone surgical interventions other than AD or TS were also excluded from the study. The surgical procedure was not decided randomly; AD was selected for power improvement and

TS for maintaining ROM, depending on whether the patient's requirements were more strongly for restoration of strength or preservation of ROM. We also recruited healthy volunteers with no radiological signs of OA as controls.

## Surgical technique

All patients were treated by a single surgeon (author A.K.) in the same institution. AD used an arthroscope to remove the articular surface of the TMC joint, followed by placing the TMC joint at 30˚ in palmar abduction and 20˚ in radial abduction. The thumb position was adjusted so that the thumb pulp opposed the pulp of the index and middle fingers when pinching. Two headless compression screws (DTJ screw, Meira, Nagoya, Japan) were used for fixation.

TS was performed as previously described with some modifications [10]. We made a 2 cm long incision over the TMC joint's dorsum, completely excising the trapezium. In addition to the method reported by Kochevar et al. [10], the free end of the abductor pollicis longus (APL) tendon graft was sutured again to the flexor carpi radialis (FCR) after passing through the APL tendon itself just before passing through the first metacarpal for a stronger suspension effect.

For both surgeries, the surgeon instructed the patient to be immobilised in a thumb spica cast for 3 weeks postoperatively and to wear a removable brace for an additional 3 weeks. After cast removal, ROM rehabilitation of the thumb was performed.

## Clinical evaluation

The patients were examined preoperatively and postoperatively. Maximum contractive force by key pinch and pulp pinch using a pinch meter (OG Giken, Okayama, Japan) and grip strength using a hand dynamometer (Preston Corp., New York, NY, USA) were recorded at the examination. The Kapandji score was evaluated to assess the functional mobility of the thumb. The score ranges from 0 to 10, with higher scores indicating greater mobility and functional use of the thumb.

Alongside the Kapandji score, the ROM of the IP joint, MP joint, and radial and palmar abduction of the TMC joint were measured using a plastic goniometer (Sakai Med, Tokyo, Japan). The patients' symptoms were recorded using the Japanese version of the Disability of the Arm, Shoulder and Hand (DASH) questionnaire [11]. This is a self-reported questionnaire designed to measure the physical function and symptoms of patients suffering from upper-limb disorders. The DASH scores range from 0 (no disability) to 100 (most severe disability), with higher scores indicating greater disability. Pain score was also recorded using the Visual Analogue Scale (VAS) [12].

## 3D motion capture

The thumb's 3D kinematics were measured using a retroreflective surface-based marker method. One retroreflective marker was placed on the thumbnail, and three markers attached to a T-shaped jig were placed on the dorsal surface of each of the first proximal phalanx, first metacarpal, and third metacarpal according to a previously described method [13, 14] (Fig 1). The segment coordinate system was defined as per the International Society of Biomechanics conventions [15]. Twelve infrared cameras (Flex 3, OptiTrack, Corvallis, OR, USA) and a 3D motion capture system (VENUS 3D, Nobby Tech, Tokyo, Japan) were used to track the markers' movements at a sampling rate of 100 Hz. Participants were instructed to perform a series of circumduction movements in which the thumb passed through the adduction, extension, abduction, and flexion positions in sequence, with 1-second intervals between each movement as previously described [13]. A total of 10 trials were performed, and the ROM was assessed by averaging the central eight movements. Kuo et al. [16] evaluated the accuracy and reliability of

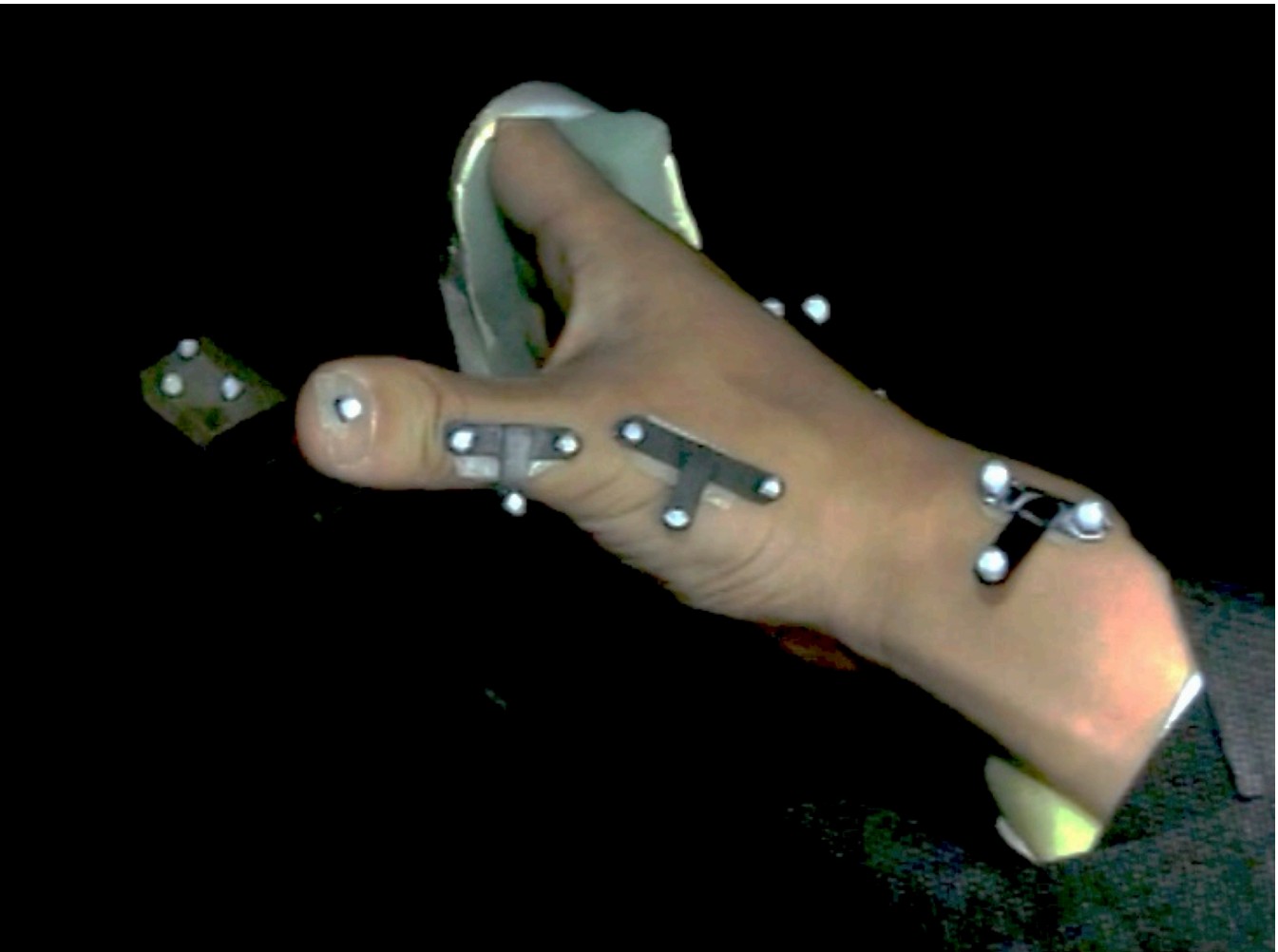

**Fig 1. Experimental setup and marker placement.**

thumb motion measurements using these surface markers with fluoroscopy and reported similarities in bone and marker motion. Therefore, we hypothesised that the skin artefacts during thumb motion had little effect on the accuracy of the surface markers and represented the motion of the underlying bone segment. At the beginning of each trial, the participants placed the thumb in a neutral position, with both abduction/adduction and extension/flexion at zero, and were instructed to fully extend the four fingers using a splint fixed from the forearm to the fingers. Furthermore, they was instructed to move the thumb in as large a circle as possible to complete the task.

## Data analysis

The path length of the tip of the thumb and the area bounded by the circumferential path of the thumb tip were calculated during circumduction. These measurements were normalised by the participant's palm width (palm correction). Palm width was measured at the line connecting the radial end of the proximal palmar crease and the ulnar end of the distal palmar crease of the hand being measured prior to motion capturing. The measurement was calculated as follows: the path length was divided by the palm width, and the area was divided by

the square of the palm width. The palm width is determined by measuring the distance between two points: the radial end of the proximal transverse palmar crease and the ulnar end of the distal transverse palmar crease. By using the palm width and its square as normalisation factors for the path length and area, respectively, we achieve a more accurate representation of the thumb's functional range, tailored to each participant's hand size.

The protocol for calculating joint angles from the marker set was based on the method previously described by Nataraj and Li [17]; joint angles were determined according to the XYZ Euler angle rotation sequence. Six rotational degrees of freedom were investigated, namely, extension/flexion, abduction/adduction, internal/external rotation of the TMC joint, extension/flexion and abduction/adduction of the MP joint, and extension/flexion of the IP joint. The angular motion of the MP and IP joints was calculated based on the change in the distal part of the joint with respect to the proximal part. The specific motion of the first metacarpal bone relative to the trapezium was not directly measured. Instead, the third metacarpal bone was used as an alternative to the trapezium. Original software created using MATLAB 2021a (MathWorks, Natick, MA, USA) was used to analyze the data. Since the measurement accuracy of our motion analysis system ranged from 0.2˚ to 0.3˚, the measurement data for motion analysis were recorded as integer values.

## Statistical analysis

All data were expressed as means and standard deviation. Tukey's multiple comparison test was performed to assess the impact of OA in healthy controls and each patient preoperatively. In addition, to compare task results and clinical values preoperatively and 1 year postoperatively, a paired t-test was performed to evaluate the effect of the surgery. Welch's t-test was performed to evaluate the differences obtained in each surgical procedure. Statistical significance was considered at p = 0.05.

## Results

Patients with TMC-OA on radiographs with Eaton stage ≥2, including 14 patients scheduled for AD and nine scheduled for TS, and nine healthy volunteers were recruited to evaluate bilateral thumb motion. The demographic characteristics of the participants are presented in Table 1. Tukey's multiple comparison test revealed significant differences in patient age between healthy volunteers and TMC-OA patients.

One year postoperatively, a radiographic union was observed in all participants who underwent AD. The statistical analysis results are shown in Tables 2 and 3.

Significant differences were observed in the clinical evaluation of VAS score, grip strength, pinch strength, DASH score, and radial and palmar abduction at the TMC (thumb metacarpal) joint between healthy volunteers and preoperative TMC-OA patients. At the MP joint, a reduced ROM in flexion was noted exclusively in the preoperative AD group.

The 3D motion analysis revealed a reduced ROM in all directions at the first metacarpal (TMC joint), as well as reduced movement in both supination and pronation at the MP joint. Additionally, the motion of the thumb tip was restricted, showing reductions in both trajectory length and area in the preoperative group.

In postoperative clinical evaluation, the VAS score significantly improved in both the AD and TS groups. Furthermore, patients in the AD group exhibited significant improvements in grip strength, pinch strength, DASH score, extension of the IP joint, and flexion of the MP joint. However, there were no improvements in clinical parameters other than the VAS score in the TS group; the DASH score showed a tendency for improvement.

**Table 1. Demographics and participant characteristics.**

|  | Healthy Volunteer (N = 18) | AD (N = 14) | TS (N = 9) |
|---|---|---|---|
| **Age (years) (Min-Max)** | 48.5 (30–60) | 68.5 (58–80) | 74.1 (62–83) |
| **Sex** |  |  |  |
| Male | 2 | 3 | 0 |
| Female | 16 | 11 | 9 |
| **Affected side** |  |  |  |
| Right | 9 | 7 | 6 |
| Left | 9 | 7 | 3 |
| **Eaton stage** |  |  |  |
| 1 | - | 0 | 0 |
| 2 | - | 1 | 0 |
| 3 | - | 13 | 7 |
| 4 | - | 0 | 2 |

Values are presented as mean with age range or n.

**Table 2. Differences in clinical assessment and three-dimensional motion analysis between healthy volunteers and preoperative TMC-OA patients.**

|  |  |  | Healthy | preop. | | p-value | | |
|---|---|---|---|---|---|---|---|---|
|  |  |  |  | AD | TS | H-AD | H-TS | AD-TS |
| Clinical assessment |  | Pain at rest (VAS) | 0 | 55(27) | 50(27) | <0.01 | <0.01 | 0.36 |
|  |  | Pain on activity (VAS) | 0 | 77(28) | 76(24) | <0.01 | <0.01 | 0.98 |
|  |  | Grip strength (kg) | 29.6(11.2) | 17.9(12.6) | 16.3(6.2) | <0.05 | <0.05 | 0.94 |
|  |  | Key pinch (kg) | 7.6(2.0) | 5.0(3.0) | 4.6(1.5) | <0.05 | <0.05 | 0.92 |
|  |  | Tip pinch (kg) | 6.8(2.1) | 3.3(2.4) | 3.8(1.2) | <0.01 | <0.01 | 0.84 |
|  |  | DASH score | 0(0) | 35.3(18.8) | 41.6(15.6) | <0.01 | <0.01 | 0.53 |
|  |  | Kapandji score | 10(0) | 9.6(0.9) | 10.0(0.0) | 0.09 | 1.00 | 0.17 |
|  | IP joint | Extension (°) | 15(8.6) | 16(11) | 18(10) | 0.97 | 0.75 | 0.88 |
|  |  | Flexion (°) | 63.1(10.9) | 57(15) | 61(9) | 0.35 | 0.92 | 0.71 |
|  | MP joint | Extension (°) | 13.1(9.3) | 10(15) | 12(11) | 0.78 | 0.99 | 0.90 |
|  |  | Flexion (°) | 58.3(9.4) | 44(16) | 50(9) | <0.05 | 0.24 | 0.45 |
|  | TMC joint | Palmar abduction (°) | 59.4(7.5) | 43(10) | 42(9) | <0.01 | <0.01 | 0.91 |
|  |  | Radial abduction (°) | 61.7(7.1) | 45(12) | 41(11) | <0.01 | <0.01 | 0.65 |
| 3D motion analysis | 1st MC | Flex/Ext (°) | 50(11) | 24(7) | 21(5) | <0.01 | <0.01 | 0.74 |
|  |  | Abd/Add (°) | 43(9) | 29(9) | 29(8) | <0.01 | <0.01 | 0.99 |
|  |  | Pro/Sup (°) | 54(14) | 21(8) | 20(8) | <0.01 | <0.01 | 0.96 |
|  | MP | Flex/Ext (°) | 67(13) | 60(13) | 64(16) | 0.30 | 0.85 | 0.75 |
|  |  | Abd/Add (°) | 39(9) | 32(9) | 33(3) | 0.07 | 0.16 | 0.99 |
|  |  | Pro/Sup (°) | 32(8) | 19(7) | 23(8) | <0.01 | <0.05 | 0.47 |
|  | IP | Flex/Ext (°) | 56(13) | 54(17) | 50(10) | 0.93 | 0.61 | 0.82 |
|  | Tip trajectory | Path length (mm) | 360(33) | 296(38) | 292(43) | <0.01 | <0.01 | 0.99 |
|  |  | Path length (%Palm width) | 4.5(0) | 3.6(0.5) | 3.7(0.5) | <0.01 | <0.01 | 0.61 |
|  |  | Area (mm$^2$) | 7722(1580) | 4906(1085) | 4584(1162) | <0.01 | <0.01 | 0.84 |
|  |  | Area (%Palm width$^2$) | 1.2(0) | 0.72(0.2) | 0.76(0.2) | <0.01 | <0.01 | 0.92 |

VAS: Visual analogue scale; DASH: Disability of the Arm, Shoulder, and Hand score; MP: metacarpophalangeal; IP: interphalangeal; TMC: trapeziometacarpal; 1st MC: first metacarpal bone; preop.: preoperation; AD: arthrodesis; TS: trapeziectomy with suspentionplasty; Flex: flexion; Ext: extension; Abd: abduction; Add: adduction; Pro: pronation; Sup: supination; Path length: the length of thumb tip trajectory path; Area: the area enclosed by the perimeter path of the thumb tip.
Values other than estimates are typically presented as the mean (with standard deviation).

**Table 3. Change values of the three-dimensional motion analysis pre- and postoperatively.**

| | | AD | | | significance (preop. - 1year) | TS | | | significance (preop. - 1year) | significance (AD—PS) |
|---|---|---|---|---|---|---|---|---|---|---|
| | | preop. | 1 year | diff. | p-value | preop. | 1 year | diff. | p-value | p-value |
| Pain at rest (VAS) | | 55(27) | 7(18) | -51(30) | <0.01 | 50(27) | 8(18) | -41(20) | <0.01 | 0.51 |
| Pain on activity (VAS) | | 77(28) | 28(24) | -44(23) | <0.01 | 76(24) | 17(27) | -58(46) | <0.05 | 0.48 |
| Grip strength (kg) | | 17.9 (12.6) | 23.4 (12.4) | 4.6(7.6) | 0.05 | 16.3(6.2) | 15.8 (4.0) | -0.5 (6.2) | 0.82 | 0.10 |
| Key pinch (kg) | | 5.0(3.0) | 6.5(2.4) | 1.5(2.2) | <0.05 | 4.6(1.5) | 4.7(1.1) | 0.1(0.9) | 0.85 | <0.05 |
| Tip pinch (kg) | | 3.3(2.4) | 5.7(2.3) | 3.9(2.2) | <0.01 | 3.8(1.2) | 3.6(1.0) | -0.2 (1.2) | 0.67 | <0.01 |
| DASH score | | 35.3 (18.8) | 16.2 (15.8) | -19.1 (16.4) | <0.01 | 41.6 (15.6) | 29.3 (27.4) | -9.8 (21.3) | 0.23 | 0.31 |
| Kapandji score | | 9.6(0.9) | 8.9(1.4) | -0.6(1.8) | 0.2 | 10.0(0.0) | 9.8(0.7) | -0.2 (0.7) | 0.35 | 0.43 |
| IP joint | Extension (°) | 16(11) | 24(13) | 8(12) | <0.05 | 18(10) | 17(10) | -1(10) | 0.73 | 0.07 |
| | Flexion (°) | 57(15) | 60(13) | 3(11) | 0.26 | 61(9) | 57(9) | -4(7) | 0.12 | 0.06 |
| MP joint | Extension (°) | 10(15) | 14(11) | 3(14) | 0.39 | 12(11) | 14(15) | 1(10) | 0.69 | 0.72 |
| | Flexion (°) | 44(16) | 48(17) | 4(6) | <0.05 | 50(9) | 43(11) | -7(14) | 0.16 | 0.06 |
| TMC joint | Palmar abduction (°) | 43(10) | 45(12) | 1(12) | 0.67 | 42(9) | 48(10) | 6(18) | 0.36 | 0.52 |
| | Radial abduction (°) | 45(12) | 49(13) | 4(15) | 0.29 | 41(11) | 54(14) | 13(19) | 0.07 | 0.25 |

VAS: Visual analogue scale; DASH: Disability of the Arm, Shoulder, and Hand score; MP: metacarpophalangeal; IP: interphalangeal; preop.: preoperation; diff.: difference between preoperative and 1-year postoperative values.

Values other than estimates are typically presented as the mean (with standard deviation).

Comparing the change in the AD and TS groups at 1 year postoperatively, the improvement in pinch force was significantly greater in the AD group than in the TS group.

The 3D motion analysis showed that the first metacarpal movement decreased in all directions, and the ROM of the MP joint decreased in extension-flexion and supination-pronation 1 year postoperatively in the AD group, whereas there was no change in the TS group (Table 4).

Comparison of the change in ROM preoperatively and 1 year postoperatively showed significant differences in extension-flexion and supination-pronation of the first metacarpal. Furthermore, there was no postoperative change in the movement of the thumb tip in the AD group, whereas, in the TS group, the area bounded by the thumb tip motion increased, but the significant difference disappeared when normalised by the palmar width. The thumb tip path trajectory in the AD group moved to the palmer direction (Fig 2).

## Discussion

This study demonstrates that the pre- and postoperative ROM in TMC-OA patients was smaller than that of healthy individuals. Both surgical procedures resulted in postoperative pain reduction and DASH score improvement. The results of the 3D motion analysis revealed that AD led to a decrease only in the flexion-extension ROM of the TMC joint, and TS did not contribute to a significant increase in ROM. Additionally, no significant change in thumb tip trajectory was noted following surgical intervention in either procedure.

In this study, the surgical procedures were performed by a single surgeon. Although there may be concerns regarding the generalisability of our findings due to this factor, it should be

**Table 4. Results of the three-dimensional motion analysis.**

| | | AD | | | significance (preop. - 1year) | TS | | | significance (preop. - 1year) | significance (AD—TS) |
|---|---|---|---|---|---|---|---|---|---|---|
| | | preop. | 1 year | diff. | *p*-value | preop. | 1 year | diff. | *p*-value | *p*-value |
| 1st MC | Flex/Ext (°) | 24(7) | 13(2) | -10.8(6.6) | <0.01 | 21(5) | 23(6) | 1.3(5.5) | 0.51 | <0.01 |
| | Abd/Add (°) | 29(9) | 22(5) | -6.9(10.0) | <0.05 | 29(8) | 28(6) | -0.5(7.0) | 0.85 | 0.11 |
| | Pro/Sup (°) | 21(8) | 16(7) | -5.3(7.3) | <0.05 | 20(8) | 17(4) | -2.5(10.4) | 0.49 | <0.05 |
| MP | Flex/Ext (°) | 60(13) | 66(13) | 6.7(11.1) | <0.05 | 64(16) | 66(12) | 1.6(8.5) | 0.58 | 0.25 |
| | Abd/Add (°) | 32(9) | 32(13) | -0.6(6.4) | 0.71 | 33(3) | 31(4) | -1.9(5.7) | 0.35 | 0.64 |
| | Pro/Sup (°) | 19(7) | 24(8) | 4.7(8.1) | <0.05 | 23(8) | 25(5) | 2.1(8.2) | 0.47 | 0.46 |
| IP | Flex/Ext (°) | 54(17) | 52(19) | -2.1(16.6) | 0.65 | 50(10) | 58(8) | 7.5(13.3) | 0.13 | 0.16 |
| Tip trajectory | Path length (mm) | 296(38) | 295(64) | 0.34(57) | 0.99 | 292(43) | 310(43) | 18.1(34.6) | 0.13 | 0.34 |
| | Path length (%Palm width) | 3.6(0.5) | 3.6(1) | 0.006 (0.703) | 0.98 | 3.7(0.5) | 4(1) | 0.240 (0.453) | 0.15 | 0.39 |
| | Area (mm$^2$) | 4906 (1085) | 4922 (210) | 16.4 (1849.9) | 0.97 | 4584 (1162) | 5425 (148) | 841(1264) | <0.05 | 0.25 |
| | Area (%Palm width$^2$) | 0.72(0.2) | 0.72(0) | 0.001 (0.282) | 0.99 | 0.76(0.2) | 0.9(0) | 0.14(0.22) | 0.08 | 0.21 |

1st MC: first metacarpal bone; MP: metacarpophalangeal joint; IP: interphalangeal joint; Flex: flexion; Ext: extension; Abd: abduction; Add: adduction; Pro: pronation; Sup: supination; Path length: the length of thumb tip trajectory path; Area: the area enclosed by the perimeter path of the thumb tip; preop.: preoperation; diff.: difference between preoperative and 1-year postoperative values.

Values except for statistical values are typically presented as the mean (with standard deviation).

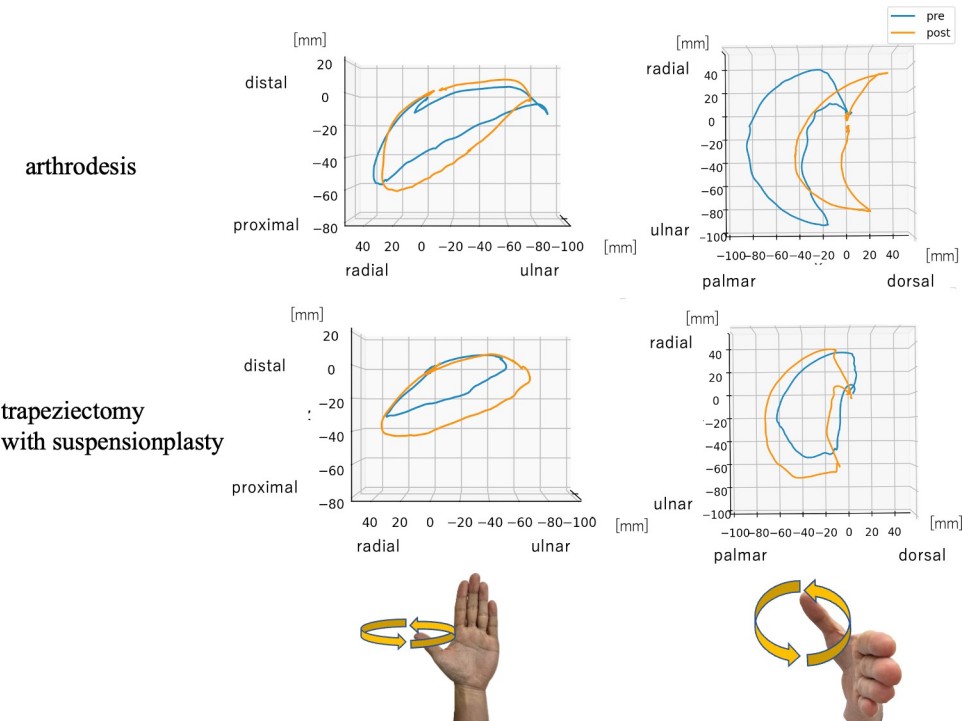

**Fig 2. Typical thumb tip path trajectories in each plane.** Upper row: arthrodesis, lower row: trapeziectomy with suspensionplasty with the position of the start of the circumduction movement as the origin. The blue line shows the preoperative trajectory, while the orange shows the postoperative trajectory.

noted that the surgeries were performed using widely accepted instruments and followed a standardised procedure that is not dependent on the specific surgeon, allowing for the potential reproduction of similar techniques by other surgeons. Moreover, the surgeon in question has over 10 years of experience in hand surgery, ensuring stability and consistency in surgical techniques. We believe this contributes to reducing variability in treatment outcomes and enhances the internal validity of our research. To mitigate the limitations imposed on the validity and generalisability of our findings by our study approach, we provide a detailed protocol of the surgical technique. This facilitates the reproduction and further evaluation of our findings in diverse clinical settings and by different surgeons. In future research, comparing treatment outcomes among different surgeons will become feasible, thereby potentially increasing the generalisability of our findings.

Omokawa et al. [18] reported a reduction in thumb circumduction motion to 25% after AD in a normal cadaver study, similar to that reported by Kawano et al. [5]. The reason for the discrepancy with our results could be that the ROM was not reduced in cadavers due to pain, and the cadavers used had healthy TMC joints, which did not decrease the preoperative ROM. Our results are considered more reliable because they are not the result of a cadaveric simulation, but of an actual patient's active motion.

The results of the 3D motion analysis showed that motion around the TMC joint remained despite AD since the TMC motion was measured based on the positional relationship between the first and third metacarpals, whose motion includes not only the TMC joint but also the scaphotrapezialtrapezoidal (STT) and radioscaphoid (RS) joints. This suggests that the TMC motion following AD was attributed to the compensatory motion of the STT and RS joints.

Koff et al. [19] performed kinematic tests using electromagnetic sensors on three different TS procedures in a cadaver study to evaluate the radius of joint motion (position of the centre of rotation) and working area. The procedure in which the centre of joint rotation is shifted closer to the articular surface of the thumb metacarpal reduced the effective radius of the joint, and the work area of the thumb reduced. Considering the centre of rotation of the metacarpal, AD shifts the centre of rotation towards the STT joint due to the fusion of the metacarpal and trapezium bones; an increased lever arm could maintain the working area of the thumb. This corresponds with the structural support reported by Koff et al. The aforementioned factors explain why, in our study, the path length and trajectory area of the thumb tip were preserved despite a reduction in the postoperative ROM after AD.

Although we expected that TS would improve the ROM and thumb tip trajectory in the postoperative period compared to the preoperative period, the ROM remained unchanged. The TS technique used in the present study shortened the lever arm after trapeziectomy and moved the centre of rotation of the first metacarpal to the metacarpal base by means of a tendon graft passed from the base of the first metacarpal to the second metacarpal. In addition, the tendon graft fixed between the FCR and APL may have increased the constraint on the first metacarpal.

Based on our investigation, when selecting the surgical technique, AD might be indicated when there is no preoperative ROM complaint and there is little risk of OA in adjacent joints. In addition, AD is expected to improve pinch strength. On the other hand, TS may be a more appropriate option in cases where the risk of OA in adjacent joints is a concern, or in patients who avoid limitation of thumb extension and adduction, or inability to lay the palm flat on a table, which is considered a weakness of AD.

This study had some limitations. First, the sample size was relatively small, and a larger sample size analysis might have revealed additional kinematic changes in both procedures. The short follow-up period of 1 year may change the results in the long term. The selection of AD or TS was not blinded; rather, it was determined by a single surgeon who performed all

procedures, which might have led to a selection bias. Finally, the ROM of the TMC joint measured via 3D motion analysis also included the motion of the STT and RS joints since it was evaluated based on the motion of the first metacarpal bone with respect to the third metacarpal bone. A detailed evaluation of the movement of each joint using computed tomography will provide deeper insights into the relationships between the TMC, STT, and RS joints.

In conclusion, we detected changes in the ROM and motor function following AD or TS for TMC-OA using 3D motion analysis. Pain was improved by both methods, and the ROM of the TMC joint was unchanged pre- and postoperatively. Understanding the kinematic change due to surgical intervention may help surgeons to better decide on the most suitable surgical method.

## Supporting information

**S1 File. Data of demographics and clinical evaluation.** M: male; F: Female; VAS: Visual analogue scale; DASH: Disability of the Arm, Shoulder, and Hand score; MP: metacarpophalangeal; IP: interphalangeal; preop.: preoperation; AD: arthrodesis; TS: trapeziectomy with suspentionplasty; Flex: flexion; Ext: extension.
(XLSX)

**S2 File. Data of demographics and three-dimensional motion analysis.** M: male; F: Female; MP: metacarpophalangeal; IP: interphalangeal; TMC: trapeziometacarpal; 1st MC: first metacarpal bone; preop.: preoperation; AD: arthrodesis; TS: trapeziectomy with suspentionplasty; Flex: flexion; Ext: extension; Abd: abduction; Add: adduction; Pro: pronation; Sup: supination; Path length: the length of thumb tip trajectory path; Area: the area enclosed by the perimeter path of the thumb tip.
(XLSX)

## Acknowledgments

The authors would like to express my profound gratitude to the Biostatistics Unit at the Clinical Research Center in Hiroshima, Hiroshima University Hospital, for their invaluable statistical consultation and support throughout this research. Their expertise and insights were instrumental in the analysis and interpretation of our data, greatly contributing to the quality and integrity of our work.

## Author Contributions

**Conceptualization:** Akira Kodama.

**Data curation:** Teruyasu Tanaka, Akira Kodama, Hiroshi Kurumadani.

**Investigation:** Teruyasu Tanaka, Kaguna Tanimoto, Shigeki Ishibashi, Masaru Munemori.

**Methodology:** Akira Kodama.

**Project administration:** Akira Kodama.

**Software:** Hiroshi Kurumadani.

**Supervision:** Akira Kodama, Nobuo Adachi.

**Writing – original draft:** Teruyasu Tanaka.

**Writing – review & editing:** Akira Kodama, Toru Sunagawa, Nobuo Adachi.

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
