## [Decision Letter · Decision Letter 0]

30 Nov 2023

PONE-D-23-33334Three-Dimensional Motion Analysis of Pre- and Postoperative Thumb Movement in Trapeziometacarpal Joint Osteoarthritis - Comparison of Arthrodesis and Trapeziectomy with SuspensionplastyPLOS ONE

Dear Dr. KODAMA,

Thank you for submitting your manuscript to PLOS ONE. After careful consideration, we feel that it has merit but does not fully meet PLOS ONE’s publication criteria as it currently stands. Therefore, we invite you to submit a revised version of the manuscript that addresses the points raised during the review process.

Further to the reviewers' comments and suggestions for improvement, I would strongly encourage the authors to consults a biostatistician with regards to their analyses.==============================

We look forward to receiving your revised manuscript.

Kind regards,

Stergios Makris

Academic Editor

PLOS ONE

4. We note that Figures 1 and 2 in your submission contain copyrighted images. All PLOS content is published under the Creative Commons Attribution License (CC BY 4.0), which means that the manuscript, images, and Supporting Information files will be freely available online, and any third party is permitted to access, download, copy, distribute, and use these materials in any way, even commercially, with proper attribution. For more information, see our copyright guidelines: http://journals.plos.org/plosone/s/licenses-and-copyright.

1. You may seek permission from the original copyright holder of Figures 1 and 2 to publish the content specifically under the CC BY 4.0 license.

Reviewers' comments:

Reviewer's Responses to Questions

**Comments to the Author**

1. Is the manuscript technically sound, and do the data support the conclusions?

Reviewer #1: Yes

Reviewer #2: Partly

2. Has the statistical analysis been performed appropriately and rigorously? 

Reviewer #1: Yes

Reviewer #2: No

3. Have the authors made all data underlying the findings in their manuscript fully available?

Reviewer #1: Yes

Reviewer #2: Yes

4. Is the manuscript presented in an intelligible fashion and written in standard English?

Reviewer #1: No

Reviewer #2: No

5. Review Comments to the Author

Reviewer #1: Your work has been carefully evaluated. This article is clinical research. Your work related with Quantifying thumb kinematic changes following TMC-OA surgery and help select surgical procedures.

The authors explained about weak point and strong point of each surgical procedure. However,the sample size was relatively small, and a larger sample size analysis might have revealed additional kinematic changes in both procedures. This manuscript should prove English from native speaker. This paper is moderate impact. So, It is sufficient quality to be able to consider for publication.

Reviewer #2: This study quantified and compared the differences in the range of motion (ROM) between the two procedures (AD and TS) as well as before/after the surgery in patients with TMC-OA. The authors summarized the characteristics of each procedure and provided the information on the selection of surgical procedure. However, it requires major revisions before publication.

General comment:

The statistical analyses throughout are unclear and likely not appropriate. All the data analyses should be reviewed by a biostatistician before resubmission.

Major comments:

Introduction

In line 64-66 on Page 4: Nevertheless, the actual effects on joint movement and the degree of improvement in motor function and its mechanisms remain unclear.

Please provide a more specific explanation of the reason why you think such that.

In line 67-69 on Page 4: This study aimed to quantify the postoperative kinematic changes between AD and TS in patients who have undergone surgery, and to clarify the characteristics of kinematic changes on using each surgical procedure. Please clarify the position or role of the arthrodesis (AD) in the context of TMC-OA.

Materials and Methods

In line 91: All patients were treated by a single surgeon (author A.K.) in the same institution.

All patients were treated by a single surgeon in this study. This raises concerns about the generalizability of the research results if there are variations in treatment outcomes based on the surgeon. Please explain your thoughts on how this aspect might impact the generalizability of the study findings.

The two-way ANOVA used in Tables 2 and 3 did not appear to adjust the baseline value of the ROM.

I have a concern about the comparison of the ROC between the two procedures and between the preoperative and postoperative stages have been performed based on the two-way ANOVA. I was also not sure in which analysis the Tukey method is used. The authors should obtain consultation from a biostatistician to assure the validity of the statistical analysis.

Moreover, in Table 2 and 3, I wonder about SD, F value, and Eta being zero. If these rounds to zero, it should be expressed as <0.1 or <0.01 instead. In addition, the reason for using these indices is unclear. For example, although the authors calculated the eta-squared, the discussion and conclusion was not based on this index.

In Table 3, did the ANOVA analysis conducted include healthy participants? If so, how were healthy participants compared to participants who underwent other procedures? The analyses are not particularly clear, so please provide a more detailed explanation in the Statistical Analysis section.

Discussion

In regards to the Discussion, as mentioned above, it appears that appropriate statistical analysis has not been conducted; therefore, there is a possibility that the surgical effects have not been adequately evaluated.

Minor comments:

Materials and Methods

In line 73: This prospective case-control study was conducted at a single institution.

This study is not conducted based on a case-control study design.

In line 109-113: The Kapandji score was evaluated and ROM of the IP joint, metacarpophalangeal (MP) joint, and radial and palmar abduction of the TMC joint were measured using a plastic goniometer (Sakai Med, Tokyo, Japan). The patients’ symptoms were recorded using the Japanese version of the Disability of the Arm, Shoulder and Hand questionnaire (DASH-JSSH ) [113 11].

Please provide more detailed explanations about the scores, for example, range and interpretations of higher score.

In line 127, 216, 244, and 251, please correct 'range of motion' to 'ROM'.

In line 144-146: The measurement was calculated as follows: the path length was divided by the palm width, and the area was divided by the square of the palm width.

Has the normalized method used in this study been applied in other studies? If so, please cite the reference. Otherwise, please elaborate the methods.

In line 161: All data are expressed as means and standard deviations (SD).

Please correct 'are' to 'were'.

In line 161: A two-way analysis of variance (ANOVA) was performed to compare task results and clinical values before and one year after surgery, also taking into account the influence of the procedure.

Please add the explanation of variables included in the two-way ANOVA.

In line 166: Additionally, we calculated the effect size using Eta-squared (η^2) to assess the magnitude of factor and interaction effects.

Please correct 'factor' to 'main'.

In line 173-174: There were significant differences in patient age between healthy volunteers and patients.

Please clarify the statistical hypothesis testing used.

In Tables 2 and 3, please explain what the bold text signifies.

In line 181, 184, 185, 214, 223, and 229, please correct 'main effect' to 'significant main effect'.

In line 196: Values other than statistical values are typically represented as the mean (with standard deviation).

Please correct 'statistical values' to 'estimates'.

6. PLOS authors have the option to publish the peer review history of their article (what does this mean?). If published, this will include your full peer review and any attached files.

Reviewer #1: No

Reviewer #2: No

---

## [Author Response · Author response to Decision Letter 0]

27 Feb 2024

1.Please ensure that your manuscript meets PLOS ONE’s style requirements, including those for file naming. 

[Response]

Thank you for your comments. Upon resubmission, we have reviewed the revised manuscript in reference to the style template to ensure it aligns with the guidelines and made necessary corrections as required.

3. We note that you have indicated that data from this study are available upon request. PLOS only allows data to be available upon request if there are legal or ethical restrictions on sharing data publicly.

 [Response]

We are grateful for this kind comment and suggestion. we will upload an anonymized dataset as a supporting information file. We have also mentioned this in the cover letter.

4. We note that Figures 1 and 2 in your submission contain copyrighted images. All PLOS content is published under the Creative Commons Attribution License (CC BY 4.0), which means that the manuscript, images, and Supporting Information files will be freely available online, and any third party is permitted to access, download, copy, distribute, and use these materials in any way, even commercially, with proper attribution. 

[Response]

Thank you for your valuable feedback. Both Figs 1 and 2 used in the manuscript are original creations by the authors, and do not infringe upon the copyright of others. For Fig 2, we have replaced the illustrated hand with a photograph in the resubmission. 

General comment:

・　The statistical analyses throughout are unclear and likely not appropriate. All the data analyses should be reviewed by a biostatistician before resubmission.

[Response]

We appreciate the reviewer’s suggestions. We have requested a review by a biostatistician as suggested and have made the following changes to the statistical methods.

In the previous submission, a pre- and post-operative 2-way ANOVA analysis was performed in the healthy, arthrodesis (AD) and arthroplasty (TS) groups, but in this resubmitted manuscript, a Paired T test was used to compare the pre- and post-operative results in each group. In addition, the Welch's T-test was used to evaluate the comparison of the difference between the difference amounts of the items obtained in each surgery.

The tables in the manuscript and the description of the results have therefore been changed.

We believe that this makes it possible to more clearly describe the differences between the changes in the techniques.

Major comments:

Introduction

・　In line 64-66 on Page 4: Nevertheless, the actual effects on joint movement and the degree of improvement in motor function and its mechanisms remain unclear.

Please provide a more specific explanation of the reason why you think such that.

[Response]

We thank the reviewer for these insightful comments. 

As noted in manuscript, Kawano et al. and Li et al. have reported a reduction in the thumb tip motion and abduction angle after AD, suggesting that joint mobility may be impaired. However, Haman et al. reported that range of motion is already reduced in TMC-OA patients, so it is unclear whether the actual reduction in range of motion due to surgery suffered by patients is significant.

Also in TS, Hatipoğlu et al. reported a decrease in postoperative mobility, whereas Wolf et al. and Schröder's study found that ROM was comparable to that of the healthy side, indicating that the results in TS are conflicting and require further investigation.

Furthermore, the paucity of studies that have specifically observed pre- and postoperative dynamic changes contributes to this ambiguity.

The manuscript has been revised in light of the above. (line 60 - 69)

In this regard, we believe that the understanding of the changes resulting from surgical intervention in TMC-OA is insufficient at present.

・　In line 67-69 on Page 4: This study aimed to quantify the postoperative kinematic changes between AD and TS in patients who have undergone surgery, and to clarify the characteristics of kinematic changes on using each surgical procedure. Please clarify the position or role of the arthrodesis (AD) in the context of TMC-OA.

[Response]

As suggested, we have added the article about the role and position of arthrodesis (AD) in TMC-OA in the Introduction Section. In addition, a description of Trapeziecctomy with Suspentionplasty (TS) has also been added (line 72-77).

Materials and Methods

・　In line 91: All patients were treated by a single surgeon (author A.K.) in the same institution.

All patients were treated by a single surgeon in this study. This raises concerns about the generalizability of the research results if there are variations in treatment outcomes based on the surgeon. Please explain your thoughts on how this aspect might impact the generalizability of the study findings.

[Response]

We acknowledge this concern and appreciate the opportunity to clarify our position on this matter.

The fact that the results were not obtained from multiple centers and multiple surgeons is a limitation, but the surgeon in this study had more than 10 years of experience as a hand surgeon, and the procedure of surgeries in this study were also performed with devices commonly used and in procedures that could be reproduced by other surgeons using similar procedures. 

We believe that this could ensure consistency and control of surgical procedures, thereby reducing variation in treatment application, and strengthens the internal validity of the study.

However, we recognize the limitations this approach imposes on the external validity and generalizability of our results. To mitigate this, we have provided a detailed protocol of the treatment procedure in our manuscript, facilitating replication and further evaluation of our findings in diverse clinical settings and by different surgeons. This detailed documentation aims to enable future studies to compare outcomes across various practitioners, thereby enhancing the generalizability of our findings.

The manuscript in the discussion section has been revised in light of the above. (line 265 - 277)

・　The two-way ANOVA used in Tables 2 and 3 did not appear to adjust the baseline value of the ROM.

I have a concern about the comparison of the ROC between the two procedures and between the preoperative and postoperative stages have been performed based on the two-way ANOVA. I was also not sure in which analysis the Tukey method is used. The authors should obtain consultation from a biostatistician to assure the validity of the statistical analysis.

Moreover, in Table 2 and 3, I wonder about SD, F value, and Eta being zero. If these rounds to zero, it should be expressed as <0.1 or <0.01 instead. In addition, the reason for using these indices is unclear. For example, although the authors calculated the eta-squared, the discussion and conclusion was not based on this index.

In Table 3, did the ANOVA analysis conducted include healthy participants? If so, how were healthy participants compared to participants who underwent other procedures? The analyses are not particularly clear, so please provide a more detailed explanation in the Statistical Analysis section.

[Response]

We thank the reviewer for the careful review of the manuscript. 

As noted in our response to the General comment, we have changed our statistical methods after consulting with a biostatistician in response to your suggestion.

For each evaluation item, Tukey's multiple comparisons were performed to compare healthy subjects with the preoperative group for each operation.

A paired T-test was also performed to determine the effect of surgery on the pre- and post-operative changes in each procedure.

For changes occurring with each operation, the difference between preoperative and 1- year postoperative values were assessed using Welch's T-test.

We have revised the whole statistical analysis section based on the above content. (line 178-183)

Minor comments:

Materials and Methods

・　In line 73: This prospective case-control study was conducted at a single institution.

This study is not conducted based on a case-control study design.

[Response]

We have corrected the relevant text to a cohort study, not a case-control study within the manuscript. (line 81)

・　In line 109-113: The Kapandji score was evaluated and ROM of the IP joint, metacarpophalangeal (MP) joint, and radial and palmar abduction of the TMC joint were measured using a plastic goniometer (Sakai Med, Tokyo, Japan). The patients’ symptoms were recorded using the Japanese version of the Disability of the Arm, Shoulder and Hand questionnaire (DASH-JSSH ) [113 11].

Please provide more detailed explanations about the scores, for example, range and interpretations of higher score.

[Response]

We have included descriptions and information about the range for both the Kapandji score and the DASH score. (line 118-120, line 123-127)

・　In line 144-146: The measurement was calculated as follows: the path length was divided by the palm width, and the area was divided by the square of the palm width.

Has the normalized method used in this study been applied in other studies? If so, please cite the reference. Otherwise, please elaborate the methods.

[Response]

Details of the normalisation calculations using this method and their meaning are described in the Data analysis section of the manuscript. (line 159- 163)

・　In line 127, 216, 244, and 251, please correct 'range of motion' to 'ROM'.

・　In line 161: All data are expressed as means and standard deviations (SD).

Please correct 'are' to 'were'.

[Response]

We have corrected the word issues pointed out in the manuscript.

'range of motion' to 'ROM'　：　line 141, 263, 282

'are' to 'were'：　line 178

(The 'range of motion' in line 216 in the previous manuscript has been eliminated because the whole of the relevant section has been corrected.)

・　In line 173-174: There were significant differences in patient age between healthy volunteers and patients.

Please clarify the statistical hypothesis testing used.

[Response]

The statistical hypothesis test used was Tukey's multiple comparison method. The same is described in the manuscript. (line 189-190)

・　In line 161: A two-way analysis of variance (ANOVA) was performed to compare task results and clinical values before and one year after surgery, also taking into account the influence of the procedure.

Please add the explanation of variables included in the two-way ANOVA.

・　In line 166: Additionally, we calculated the effect size using Eta-squared (η^2) to assess the magnitude of factor and interaction effects.

Please correct 'factor' to 'main'.

・　In Tables 2 and 3, please explain what the bold text signifies.

・　In line 181, 184, 185, 214, 223, and 229, please correct 'main effect' to 'significant main effect'.

・　In line 196: Values other than statistical values are typically represented as the mean (with standard deviation).

Please correct 'statistical values' to 'estimates'.

[Response]

Thank you for pointing out these issues.

We have changed the description of the table due to a change in the statistical methodology, so we believe that the points you mentioned have been resolved.

We look forward to hearing from you regarding our submission. We would be glad to respond to any further questions and comments that you may have.

---

## [Editor Report · Decision Letter 1]

16 Apr 2024

Three-Dimensional Motion Analysis of Pre- and Postoperative Thumb Movement in Trapeziometacarpal Joint Osteoarthritis - Comparison of Arthrodesis and Trapeziectomy with Suspensionplasty

PONE-D-23-33334R1

Dear Dr. KODAMA,

We’re pleased to inform you that your manuscript has been judged scientifically suitable for publication and will be formally accepted for publication once it meets all outstanding technical requirements.

Kind regards,

Stergios Makris

Academic Editor

PLOS ONE
---

## [Editor Report · Acceptance letter]

29 Apr 2024

PONE-D-23-33334R1 

PLOS ONE

Dear Dr. Kodama, 

I'm pleased to inform you that your manuscript has been deemed suitable for publication in PLOS ONE. Congratulations! Your manuscript is now being handed over to our production team.

Kind regards, 

on behalf of

Dr. Stergios Makris 

Academic Editor

PLOS ONE